# Strain Rate Dependence of Compressive Mechanical Properties of Polyamide and Its Composite Fabricated Using Selective Laser Sintering under Saturated-Water Conditions

**DOI:** 10.3390/mi13071041

**Published:** 2022-06-30

**Authors:** Xiaodong Zheng, Jiahuan Meng, Yang Liu

**Affiliations:** 1Faculty of Mechanical Engineering & Mechanics, Ningbo University, Ningbo 315211, China; zhengshangjue@163.com (X.Z.); mengjiahuan2022@163.com (J.M.); 2Key Laboratory of Impact and Safety Engineering, Ministry of Education, Ningbo University, Ningbo 315211, China

**Keywords:** selective laser sintering, carbon fiber reinforced polyamide composites, strain rate, hygroscopicity

## Abstract

In this work, polyamide 12 (PA12) and carbon fiber reinforced polyamide 12 (CF/PA12) composites were fabricated using selective laser sintering (SLS), and the coupling effects of the strain rate and hygroscopicity on the compressive mechanical properties were investigated. The results showed that the CF/PA12 had a shorter saturation time and lower saturated water absorption under the same conditions, indicating that the SLS of CF/PA12 had lower hydrophilia and higher water resistance when compared to the SLS of PA12. It was observed that as the strain rate increased, and the ultimate compression strength and the yield strength monotonically increased with almost the same slope, indicating that the strain rate had the same positive correlation with the compressive strength of the SLS of PA12 and CF/PA12. The water immersion results showed a significant reduction of 15% in the yield strength of SLS of PA12, but not very significant in CF/PA12. This indicated that the carbon fiber was favorable for maintaining the mechanical properties of polyamide 12 after absorbing water. The findings in this work provide a basic knowledge of the mechanical properties of SLS polyamide under different loading and saturated-water conditions and thus is helpful to widen the application of SLS products in harsh environments.

## 1. Introduction

Selective laser sintering (SLS) is an important branch of laser additive manufacturing technology and an important processing method used for the fabrication of polymer products [1,2]. Compared with the traditional plastic molding processes, such as extrusion molding [3] or injection molding [4], SLS is capable of fabricating complex and fine plastic workpieces with less loss of raw material, because the recycled raw materials can be used in the next fabrication. Thus, it can reduce production costs significantly.

Polyamide, also known as nylon, along with its composites has been the most widely used engineering plastics in recent decades due to its excellent performance, such as heat resistance, impact resistance, high strength, anti-seismic, etc. [5]. The components made from nylon are light, non-rusting, and post-maintenance, which has caused nylon parts to gradually replace some of the metal parts in automobile and consumer electronics industries [6]. As a result, the mechanical performance of polyamide became the focus of attention. Connor et al. [7] comparatively studied the tensile properties of the polyamide 12 and glass bead reinforced polyamide 12 composites, and found that the addition of glass beads increased the tensile and bending strength by 39% and 15%, respectively. Cai et al. [8] compared the tensile strength of SLS polyamide 12 in different directions and found that the difference of strength in X- and Y-directions was very small, and it exhibited approximate isotropy.

As polyamide and its composites are widely used in complex service conditions, the influence of the load strain rate on its properties is focused [9,10]. Wang et al. [11] studied the influence of the tensile strain rate on the elastoplastic deformation and failure behavior of polyamide composites. Although the strain rate had a limited effect on the deformation and failure characteristics, Young’s modulus increased significantly upon the increasing strain rate, indicating that loading strain rate had a significant influence on the mechanical properties of polyamide 12 and its composites. Sagradov et al. [12] proposed a method to analyze the strain rate-related material and damage behavior of polyamide 12 by SLS. In this work, two different situations were considered: multiple tensile tests where the strain rate changed during the tensile load; multiple relaxation tests where the strain rate changed at the same time each test.

Polyamide is a semi-crystalline thermoplastic with polar amide groups. When exposed to hydrothermal conditioning, the absorbed water molecule replaces the existing inter-chain amide-amide bonding with amide-water bonding [13]. This highly impacts the mechanical properties. There have been several studies regarding the hygrothermal behavior of glass fiber reinforced polyamides (GF/PA) composites on water diffusion and mechanical properties. Li et al. [14] found that the tensile, bending, and interlaminar shear strength of CF/PA6 composite are decreased by 35%, 53%, and 5%, respectively, after being immersed for 40 h. Lin et al. [15] immersed carbon fiber reinforced polyamide 6 samples into the water at different temperatures (20 °C, 40 °C, and 60 °C), and found that tensile strength, Young’s modulus, and impact strength decrease monotonously with the increasing temperature. Chaichanawong et al. [16] found that the mechanical properties of glass fiber reinforced polyamide after being saturated with water were highly related to the immersion time. Within the initial 35 days, the tensile strength decreased mildly and then decreased sharply as prolonging immersion time. Do et al. [17] comparatively studied the mechanical properties of polyamide-6 and polypropylene after being saturated with water. It was found that the polyamide 6 had better ultimate tensile strength, elastic modulus, and elongation than those of polypropylene.

In the research of fabric reinforced composites, SHPB (Split Hopkinson Pressure Bar) apparatus was often used for the experimental determination of dynamic mechanical properties. Yang et al. [18] analyzed the stress uniformity of split Hopkinson bar specimens. Song et al. [19] studied the compression behavior of braided carbon/epoxy laminate composites under in-plane and out-of-plane loads using the SHPB device. The results showed that the stress–strain curve, maximum compressive stress and strain all change with the strain rate.

From the literature, it is observed that the strain rate and hygroscopicity have a great influence on the mechanical properties of polyamide. However, it is observed that limited effort is made to the coupling effect of these two factors. In this work, polyamide 12 and the carbon fiber reinforced polyamide 12 composites were prepared using selective laser sintering, and then a series of compressive tests of under different strain rates (10^−4^ to 2000 /s) were conducted. Before testing, these samples were immersed into the water until they were saturated, then the influence of strain rate and hygroscopicity on the compressive properties were comparatively studied.

## 2. Experimental Detail

### 2.1. Materials

In this study, polyamide 12 (PA12) powder with spherical shape and mean particle size of 120 μm is used in this work, the apparent density is 0.48 g/cm^3^. The composite powder is prepared by mixing 20 wt.% of short carbon fiber into the polyamide 12 powder (CF/PA12) and is mixed uniformly by ball milling, the composite powder is gray-black powder with density of 0.52 g/cm^3^. A selective laser sintering apparatus (HT252P, Hunan Farsoon High-Technology Co., Ltd., Hunan, China) was employed to prepare the samples. The schematic diagram of SLS is shown in Figure 1, which is equipped with a 60 W carbon dioxide laser with a focal laser beam diameter of ≤0.5 mm. The scanning system used was a dual-axis mirror positioning system and a galvanometer optical scanner, which directs the laser beam in the X and Y axes through the F-theta lens. The building envelope was 250 × 250 × 300 mm^3^. A heater was equipped to preheat the raw powder material, which could provide a maximum temperature of up to 225 °C. During the process, high purity nitrogen was filled into the chamber to protect the sample from oxidation.

Cylinder samples with dimensions of φ8 × 8 mm are prepared by SLS, the processing parameters are determined as laser power of 45 W, laser scanning speed of 10 m/s, layer thickness of 0.1 mm. In order to investigate the influence of hygroscopicity on the mechanical properties of SLS PA12 and its composites, the as-built PA12 and CF/PA12 samples are immersed into distilled water at room temperature for 72 h, and one group of the as-built samples is used as counterpart. Then quasi-static compression and SHPB tests are performed on immersed and unimmersed PA12 and CF/PA12 samples.

### 2.2. Differential Scanning Calorimetry (DSC)

The thermal analysis of CF/PA12 and PA12 samples is carried out by differential scanning calorimeter. A sample weighing approximately 410 mg is heated from room temperature to 450 °C at a rate of 10 °C/min using argon as a protective gas. The melting temperature is determined at the maximum heat capacity and temperature. Degrees of crystallinity *Xc* are determined using DSC on Q200 equipment. In composite materials, the degree of crystallinity *Xc* is determined as follows, Equation (1):(1)$$\Delta X_{c}=\frac{\Delta H_{f}}{\Delta H_{f}^{0}\times \left(1-W_{f}\right)}$$
where Δ*H_f_* is the enthalpy of fusion of the tested polymer and Δ*H_f_^0^* the theoretical enthalpy of fusion for a 100% crystalline material and *W_f_* the fibre weight fraction. The latter was determined by TGA from matrix burn off tests at 450 °C for 1.5 h under argon (Ar).

### 2.3. Compression Tests

Quasi-static compression tests are carried out on Instron 5966 servo hydraulic material test machine with compression strain rates of 0.0001 and 0.1, respectively. At least three specimens are tested and average compressive stress–strain curves were obtained. Compression is carried out along the out-of-plane direction of the composite sample. As a contrast test, the quasi-static sample has the same geometry and size as the dynamic compression sample.

Out-of-plane compression tests are carried out on samples at high strain rates using a SHPB (Φ7, Key Laboratory of Impact and Safety Engineering, Ningbo University) apparatus. The detail principle for SHPB can refer to our previous work [20]. Different strain rates are obtained by changing the chamber pressure from 0.15 MPa to 0.45 MPa.

## 3. Results and Discussion

### 3.1. Water Absorption of SLS PA12 and CF/PA12

Under the same environmental conditions, an electronic scale with a measurement accuracy of 0.0001 g was used to test the weight of the two group samples before and after the immersion, and the water absorption rate is expressed as follows [18]:(2)$$\Delta m=\frac{m_{1}-m_{0}}{m_{0}}\times 100\%$$
(3)$$\Delta M=m_{0}-m_{2}$$

In Equations (2) and (3), Δ*m* represents the water absorption rate of material, *m*_0_ and *m*_1_ represent the weight of the sample before and after water immersion respectively. And *m*_2_ represents the weight of the sample after drying. Δ*M* represents the weight of sample hydrolysis. The above formula was used to analyze and calculate the corresponding relationship between the average water absorption of the two materials.

From Figure 2, it is observed that Δ*m* of PA12 sample increases sharply in the first 36 h as the samples were immersed into water. With further prolonged immersed time, the Δ*m* increases slowly and reaches a saturation state with Δ*m* of 5.47%. In contrast, the Δ*m* of CF/PA12 sample shows a faster upward trend in the first 24 h, and then it reaches a saturation state with Δ*m* of 4.6% once the immersion time exceeds 24 h, indicating that shorter saturation time is needed to achieve a saturation state for the CF/PA12. This is because the carbon fiber has a high specific surface area and is uniformly dispersed in the PA12 matrix, which is able to decrease the diffusion distance of water molecules in the composite.

In general, the SLS of CF/PA12 has lower hydrophilia and higher water resistance compared with PA12. This is due to the fact that, due to the higher content of the amide group and lower crystallinity, the water absorption of polyamide is better. Although the SLS of PA12 contains a lot of amide groups, the addition of carbon fiber reduces the composition, amide group content, and crystallization property of PA12 [21,22], thus reducing the water absorption rate in CF/PA12. 

Figure 3a shows the change of melting peak at different soaking times. The results show that the enthalpy of melting peak increases with the increase of immersion time. In addition, there is no change in the form of the melting peak. Then, in Figure 3b, we found that after 3 days of immersion, the crystallinity ratio of PA12 increased from 9.3% to 10.2%, and the crystallinity ratio of CF/PA12 increased from 10.7% to 11.7%. This process is associated with the phenomenon of chemical crystallization, which is well known in the literature. When chain breaking occurs, the amorphous chain in the polymer regains sufficient fluidity to form new microcrystals. The increase of crystallinity ratio has a significant effect on the mechanical properties of nylon materials. As can be seen from Figure 3b, after 3 days of aging, the crystallinity ratio increased from about 9% to about 11%. This process is associated with the chemi-crystallization phenomenon, well known in the literature [22,23]. Thus, the increase of crystallinity results in the decrease of ductility [23]. This is highly consistent with the test results in Figure 3b.

Figure 4 shows the surface morphology of PA12 and CF/PA12 before and after immersion. Figure 4a,c show that the pore size of the nylon sample increases slightly after immersion, and there are two large holes. Then, the surface of CF/PA12 in immersion quality have no obvious change, only small pore space.

### 3.2. Compressive Mechanical Properties

#### 3.2.1. Influence of Strain Rate

Figure 5 illustrates the compressive stress–strain curves of PA12 and CF/PA12 under different strain rates (10^−4^ to 2000/s). From the figure, it is observed that within the quasi-static loading range, the stress–strain curves have the same shape, and it is easy to distinguish the boundary point between the elastic stage (~5%) according to the curves. Under impact load, the elastic strain of PA12 and CF/PA12 nylon samples does not exceed 2% and 3%, respectively. Further, the stress–strain curves depend on the strain rate, and, therefore, the ultimate compression strength (UCS) and yield strength (YS) monotonically increase as the strain rate increases from 10^−4^ to 2000/s. The maximum difference of YS within the strain rate range was up to 62 MPa. Moreover, the flow stress of PA12 and CP/PA12 within the plastic stage increase with increasing strain, indicating that the PA12 and CP/PA12 have a strain strengthening effect. This kind of work-hardening ability is advantageous in structural applications to guarantee a large safety margin before fracture.

The addition of carbon fiber poses a great influence on the compressive mechanical properties of SLS PA12. The yield stress of the two groups of samples was extracted from the stress–strain curves and is presented in Figure 6. The yield stress of two groups of material almost increased linearly with logarithmic strain rate, illustrating the obvious strain rate hardening effect. Moreover, the slope of the lines is found to be almost the same, indicating that strain rate has the same influence on the PA12 and CF/PA12. 

Moreover, it is also observed that the yield stress of CF/PA12 is much higher than that of PA12 under the same strain rate (the former is 15–25 MPa higher than the latter). This is due to the reinforcement caused by carbon fiber, and the reinforcement mechanisms will be discussed later. 

It is observed that as the strain rate decreased, the reinforcing effect on yield strength and strain of carbon fiber was enhanced monotonously. Moreover, the reinforcing effect of yield strength was much larger than that of yield strain.

It is observed that the slope of the curve in Figure 7 steepens as the strain rate increases. As can be seen from Figure 7, with the increase of strain rate, the peak stress and modulus of composite material increase significantly. Although the peak strain decreases with the increase of strain rate, the dynamic failure strain and failure stress are much lower than the quasi-static failure strain.

#### 3.2.2. Influence of Water Immersion

Figure 8 depicts the compressive stress–strain curves of the PA12 at different strain rates (10^−4^ to 2000/s). It is found that in the elastic deformation stage, the influence of water immersion on the elasticity modulus is negligible. This finding is different from the melting pultrusion impregnation of PA6 [24]. Nevertheless, after yielding, the yield strength of immersed PA12 is found to be smaller than that of the unimmersed PA12 at each strain rate. Normally, the water immersion causes the compressive strength to reduce by more than 15% at different strain rates (as illustrated in Figure 9a). This result is consistent with the finding reported by [25]. This also indicates that immersion water has a negative influence on the mechanical properties of SLS PA12. The reasons may be as follows: 

The reversible hydrolysis reaction of the molecular chain takes place after absorbing water and leads to the reduction of the molecular weight of PA12 polymer. As consequence, the compressive strength of PA12 is also decreased after water immersion. Further, the amide groups occur repeatedly in the PA12 molecular chain belonging to the polar group. When the molecular chain does not contain a water molecule, the hydrogen atom on the amide group combined with the carbonyl group on another amide group to form a hydrogen bond thus increasing the crystallinity of the PA12. Meanwhile, the intermolecular force is strengthened simultaneously. However, after PA12 was soaked, a certain number of water molecules were stored in the pores of the sample. The water molecules cause the carbonyl group in the nylon molecular chain to dissociate from the hydrogen in the amide group, instead, forming a closer hydrogen bond with the water molecule. Thus, the interaction force between nylon molecules is reduced. In combination with the decreasing molecular weight caused by hydrolysis which reduces the compressive strength [24].

With regard to the SLS of CF/PA12, as shown in Figure 9, it is observed that the difference of compressive strength between immersed and unimmersed CF/PA12 is less than 4 MPa (8%), and the yield strain changes slightly, indicating that the influence of water immersion is relatively smaller than that of SLS of PA12. The reason is that after the nylon CF/PA12 samples were immersed in water, the water molecules absorbed by the sample were tightly locked by the carbon fiber, as it has strong water absorption and locking performance. Therefore, the hydrolysis reaction between the water molecules and polymer molecular chains is prevented effectively. As a consequence, the compressive mechanical properties of CF/PA12 immersed in water are slightly influenced. This indicates that the carbon fiber reinforced nylon can maintain its mechanical properties in humid environments or after being immersed in water.

Figure 10 shows the change of yield strength with the increase of strain before and after immersing of PA12 and CF/PA12. Figure 10a shows that the yield strength of PA12 before immersing is stronger than after immersing. Figure 10b shows that the yield strength of CF/PA12 is almost unaffected by the immersing factor as the strain rate increases. 

Moreover, it is found that both the strain hardening rate of PA12 and CF/PA12 decrease slightly after water immersion, and the reasons are as follows: the strain hardening behavior of polyamide after yielding is due to the orientation and crystallization of chain segments under an external force. However, the regularity of molecular chains is distorted after immersion, and, thus, causes a smaller inter-chain force between the molecular chains. Further, the orientation of chains is easy to take place, thus causing a lower strain hardening rate [13].

### 3.3. Fracture Surface

In the quasi-static compression experiment, only CF/PA12 samples were broken, so the cross-section was photographed by SEM to reveal the cause of fracture.

As shown in Figure 11a, the PA12 samples did not break under quasi-static compression. The CF/PA12 samples tend to contract inward in the vertical direction and expand outward in the horizontal direction under compression loading. Moreover, the rates of contraction or expansion are found to increase upon the increasing strain rate. This could be due to the fact that compression usually leads to inhomogeneous deformation along the compression direction and the radial direction due to the frictional force between the sample and the compression anvils [26]. 

Further, as stated before, the carbon fibers were stretched to fracture, the reasons are as follows: Under compressive deformation, the samples expand outward in the horizontal direction; however, the carbon fibers embedded in the sample were fixed by the matrix material, thus causing the fibers to stretch along the radial direction of the circle (stage II in Figure 11b). It is also observed that the core area of a compressed sample experiences the smallest deformation strain while the outer edge area is deformed most heavily [27], which causes cracks on the edge initially, thus causing the carbon fibers to fracture along the tangent of the circle ultimately (stage III in Figure 11b). It should be noted that the fracture surfaces shown in Figure 11 were prepared by cutting the fractured samples along the cracks; thus, the fracture surfaces of carbon fibers were mainly perpendicular to the observation plane.

After compressive loading, the PA12 samples were compressed into a drum shape, but not fractured at all, while CF/PA12 samples fractured, their surfaces could give valuable information on the fracture behavior and modes, as illustrated in Figure 12. The samples did not break under high-speed impact. Therefore, what we showed is the samples broken under quasi-static condition. From the figure, it is observed that within the quasi-static loading range, the fracture surfaces are rough and uneven. This kind of morphology reflects tensile rather than compression deformation characteristics of polyamide [28]. In addition, the fracture surfaces of carbon fiber also exhibit tensile characteristics, and the surface becomes more flat under a higher strain rate. Moreover, cracks always exist close to the carbon fiber, indicating that the carbon fibers were stretched relative to the matrix during compression.

## 4. Conclusions

In this work, the influences of strain rate and hygroscopicity on the compressive properties of selective laser sintering (SLS) of polyamide 12 (PA12) and the carbon fiber reinforced polyamide 12 (CF/PA12) composites were comparatively studied, and the following conclusions are drawn. 

The CF/PA12 had shorter saturation time and lower saturated water absorption under the same conditions, indicating that the SLS of CF/PA12 had lower hydrophilia and higher water resistance when compared with that of the SLS of PA12.

With the increasing strain rate, the ultimate compression strength and yield strength monotonically increased with almost the same slope, indicating that the strain rate had the same positive correlation with the compressive strength of SLS of PA12 and CF/PA12.

Compared with quasi-static state, PA12 and CF/PA12 can withstand nearly twice the yield strength under impacting load thanks to their good plasticity. Therefore, these two materials are better able to resist impact loading.

Water immersion resulted in a significant reduction of 15% in yield strength of SLS of PA12, but not so much to the CF/PA12, indicating that the carbon fibers favor for maintaining mechanical properties of polyamide 12 after absorbing water.

## Figures and Tables

**Figure 1 micromachines-13-01041-f001:**
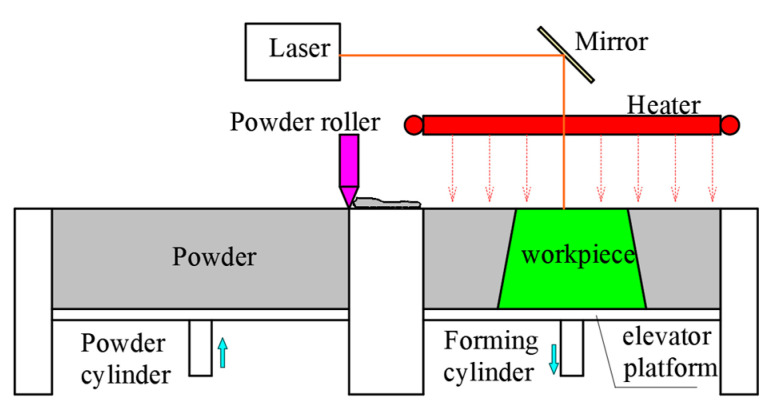
Schematic diagram of the selective laser sintering.

**Figure 2 micromachines-13-01041-f002:**
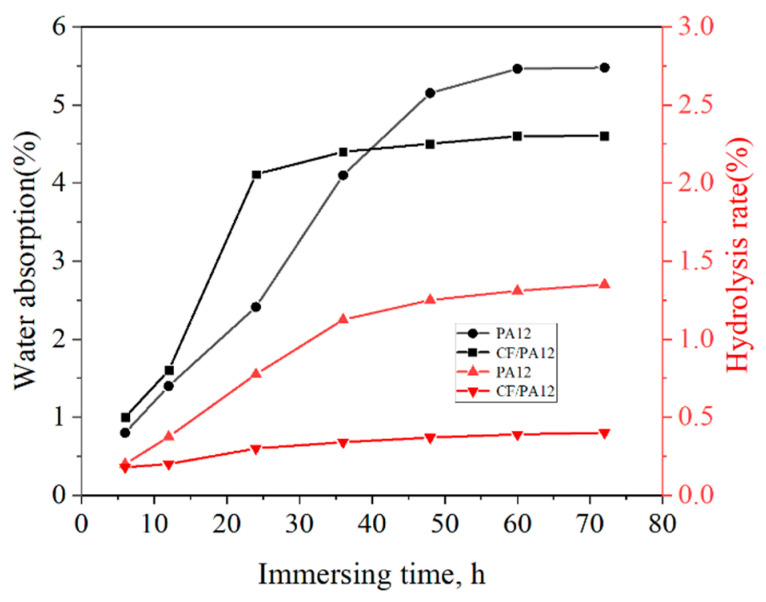
Relationship among the water absorption rate, hydrolysis rate, and immersion time.

**Figure 3 micromachines-13-01041-f003:**
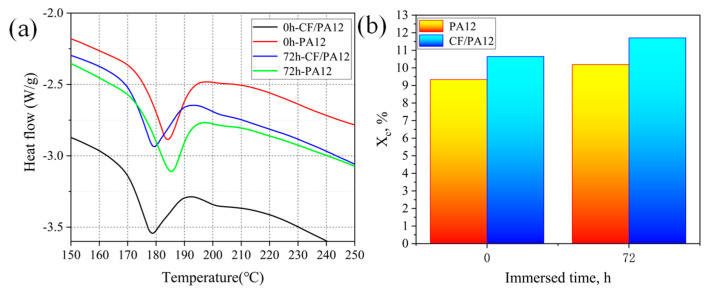
Effect of aging at water on (**a**) the melting peak and (**b**) the crystallinity ratio.

**Figure 4 micromachines-13-01041-f004:**
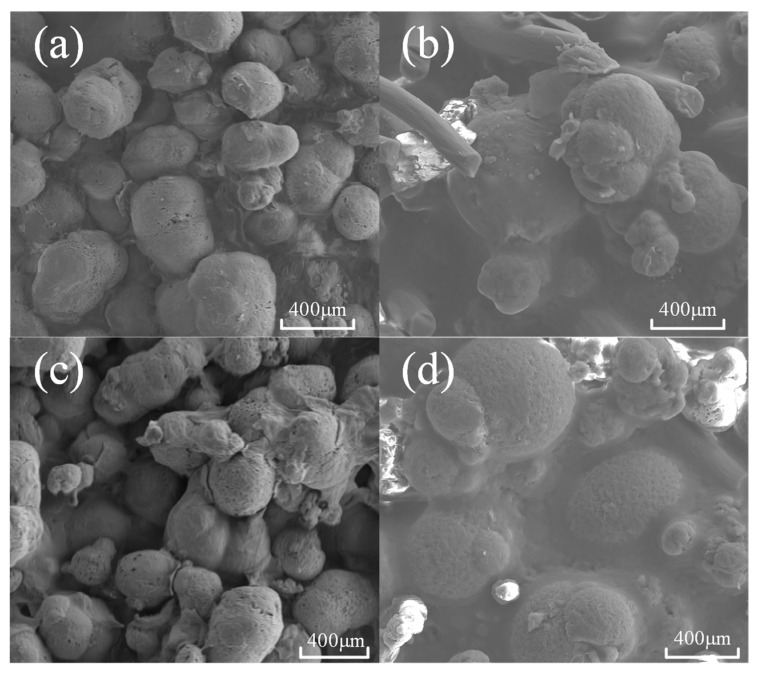
Surface morphology of CF/PA12 and PA12 before and after immersion: (**a**) 0 h-PA12; (**b**) 0 h-CF/PA12; (**c**) 72 h-PA12; and (**d**) 72 h-CF/PA12.

**Figure 5 micromachines-13-01041-f005:**
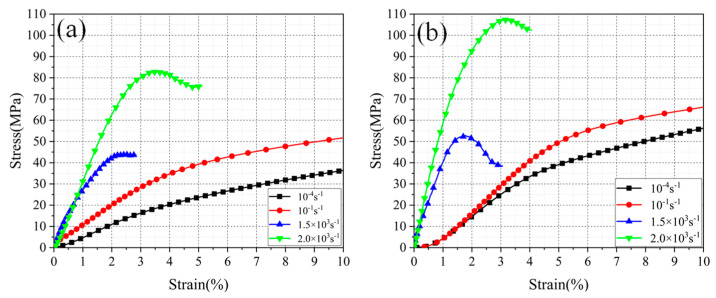
Compressive stress–strain curves of the (**a**) PA12 and (**b**) CF/PA12 under different strain rates.

**Figure 6 micromachines-13-01041-f006:**
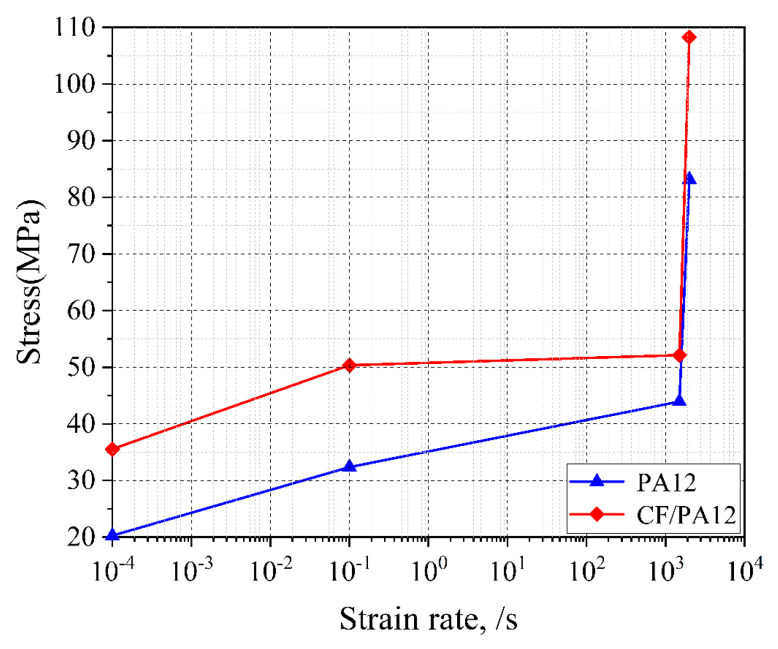
Effect of strain rate on the compressive mechanical properties of PA12 and CF/PA12 samples.

**Figure 7 micromachines-13-01041-f007:**
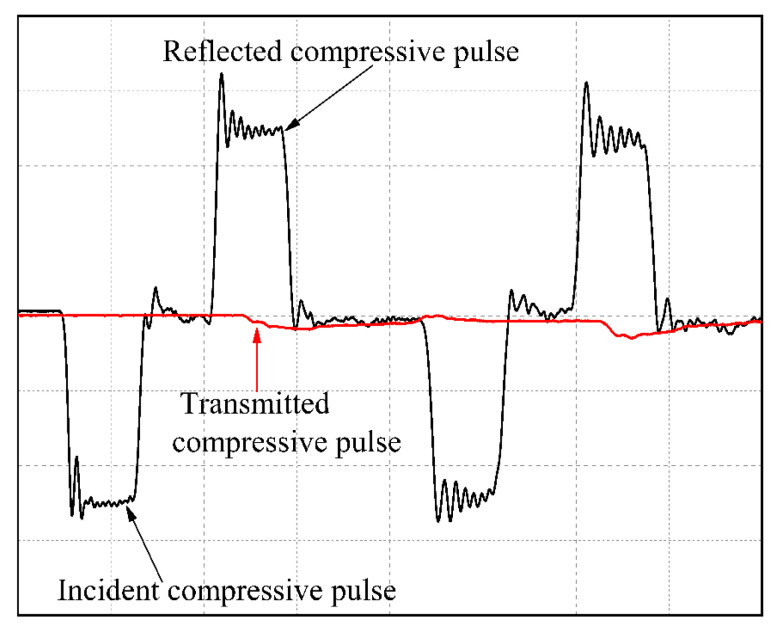
Typical dynamic compressive responses from strain gages.

**Figure 8 micromachines-13-01041-f008:**
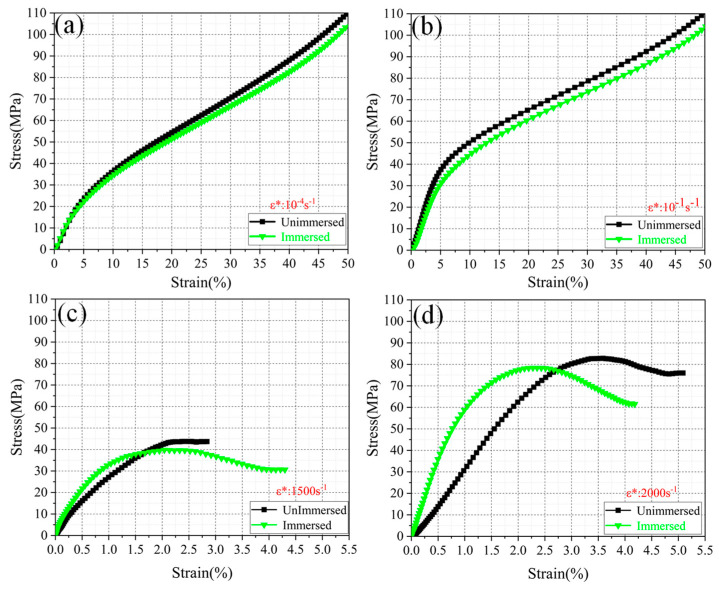
The stress–strain curves of SLS of PA12 under different strain rates: (**a**) 10^−4^ s^−1^; (**b**) 10^−1^ s^−1^; (**c**) 1500 s^−1^; (**d**) 2000 s^−1^.

**Figure 9 micromachines-13-01041-f009:**
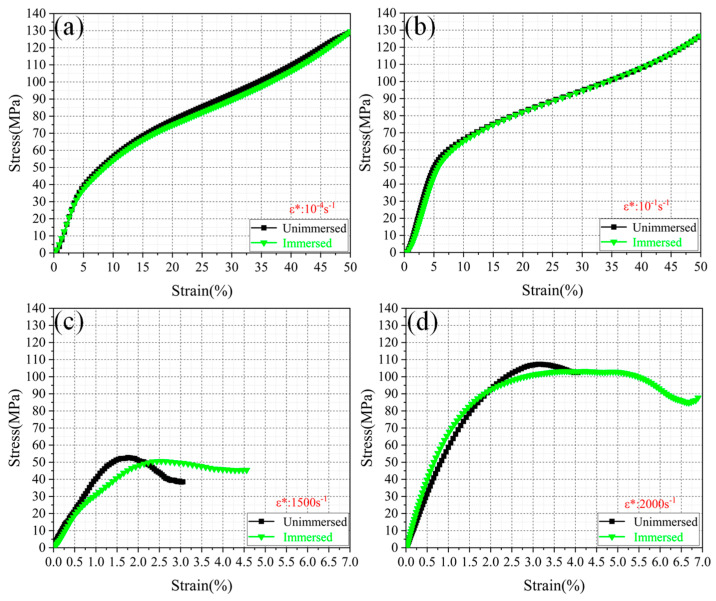
The stress–strain curves of SLS of CF/PA12 under different strain rates: (**a**) 10^−4^ s^−1^; (**b**) 10^−1^ s^−1^; (**c**) 1500 s^−1^; (**d**) 2000 s^−1^.

**Figure 10 micromachines-13-01041-f010:**
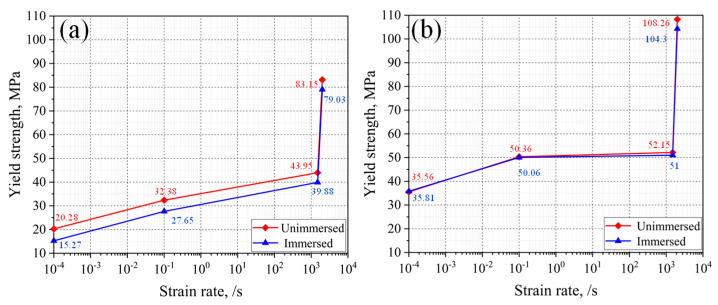
Comparison of yield strength versus strain rate of (**a**) PA12 and (**b**) CF/PA12.

**Figure 11 micromachines-13-01041-f011:**
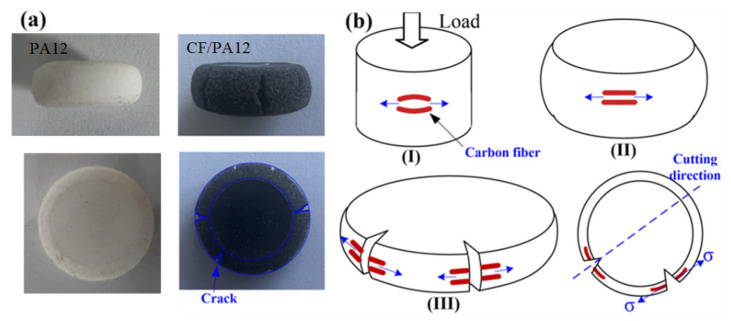
Schematic diagram for the cracking process of CF/PA12 and PA12 Sample: (**a**) PA12 and CF/PA12; (**b**) CF/PA12.

**Figure 12 micromachines-13-01041-f012:**
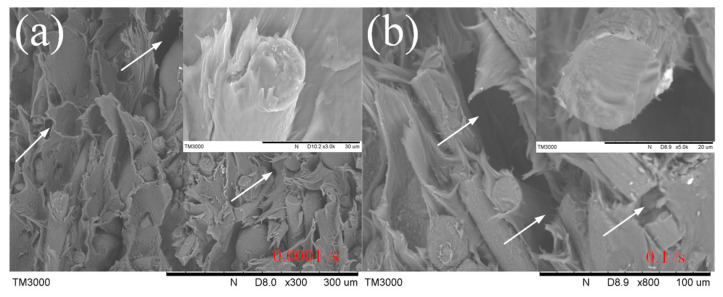
Fracture surfaces of CF/PA12 under different strain rates: (**a**) 10^−4^ s^−1^; (**b**) 10^−1^ s^−1^.

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
