# Peer review of "Strain Rate Dependence of Compressive Mechanical Properties of Polyamide and Its Composite Fabricated Using Selective Laser Sintering under Saturated-Water Conditions"

_micromachines, 2022, doi:10.3390/mi13071041_

Round 1

Reviewer 1 Report

A very interesting paper about often neglected but very important issue of long-term mechanical properties of additive manufactured parts. This is especially important when such parts will perform some long-term function in harsh (in your case wet or moist environment). I have only one minor suggestion for improving your paper. All diagrams on figures 4, 7 and 9 should have the same Y axis range (MPa), This way it will be much easier to visually compare them.  

Author Response

Point1: A very interesting paper about often neglected but very important issue of long-term mechanical properties of additive manufactured parts. This is especially important when such parts will perform some long-term function in harsh (in your case wet or moist environment). I have only one minor suggestion for improving your paper. All diagrams on figures 4, 7 and 9 should have the same Y axis range (MPa), This way it will be much easier to visually compare them.  

Response 1: Thanks for the valuable comments of reviewers, we modified the drawings in the article.

Reviewer 2 Report

This manuscript deals with the water absorption and compressive mechanical properties of PA12 reinforced with carbon fibers which were printed by the selective laser sintering process. Authors specified the properties of PA12 under the saturated conditions. Furthermore, the mechanical properties has been investigated following the different strain rate. The following comments will be helpful to modify the quality of the manuscript to be published in Micromachines.

1.     The information of the experimental equipment (e.g. model name and company) can be provided for the better understanding of readers to repeat the expeirments.

2.     Please define abbreviation “SHPB” in Introduction section.

3.     Fig. 4b in “Result and discussion” section should be revised into Figure 3b considering the context.

4.     The water absorption in the printed parts is highly related to morphology of the printed surface as SLS generally induces lots of surface pores. Thus, at least, surface SEM images should be expressed in the manuscript. And the discussion of the relationship between the surface morphology and water absorption should be added in the manuscript.

5.     The condition in Figure 4 and Section 3.2.1. should be matched (10-4 to 2000/s).

6.     The legend in Figure 3(a) should be revised. I think one of blue and green lines should be 72h-PA12.

7.     In Figure 3, the height of CF/PA12-0h was higher than pure PA12. Please explain why.

8.     The ‘space’ should be added between numbers and units.

9.     It is better to make the same range of Y-axis in Figure 4 for the direct comparison of (a) and (b).

10.  There is no explanation about Figure 6 in the manuscript.

11.  To Figure 9, the expression in figures and manuscript are different. They should be matched.

12.  In section 3.2.2., “ more than 15%” and “less than 4 MPa (8%)” should be explained how they were calculated. It will be helpful to add magnified images to explain those values.

13.  In section 3.3, it seemed better to place Figure 11 before Figure 10 for the better understanding of cross-sections.

14.  The purpose of section 3.3 is not clear. It is better to mention about the relationship of section 3.3 and related experiments between previous and following chapters.

15.  The comparison in necessary between PA12 and CF/PA12 to mention that the only CF/PA12 was fractured as authors mentioned.

Author Response

Point 1:  The information of the experimental equipment (e.g. model name and company) can be provided for the better understanding of readers to repeat the expeirments.

Response 1: The quasi-static test equipment is Instron 5966. SHPB ( Φ7, Key Laboratory of Impact and Safety Engineering, Ningbo University).

Point 2: Please define abbreviation “SHPB” in Introduction section.

Response 2: SHPB full name is Split Hopkinson Pressure Bar.

Point 3: Fig. 4b in “Result and discussion” section should be revised into Figure 3b considering the context.

Response 3: Due to the negligence of writing, this error has been corrected.

Point 4:  The water absorption in the printed parts is highly related to morphology of the printed surface as SLS generally induces lots of surface pores. Thus, at least, surface SEM images should be expressed in the manuscript. And the discussion of the relationship between the surface morphology and water absorption should be added in the manuscript.

Response 4: In this paper, we add the surface topography of two kinds of samples before and after immersion.

Figure 4 shows the surface morphology of PA12 and CF/PA12 before and after immersion. Fig. 4a and 4c show that the pore size of the nylon sample increases slightly after immersion, and there are two large holes. Then, the surface of CF/PA12 in immersion quality have no obvious change, only small pore space.

Point 5:The condition in Figure 4 and Section 3.2.1. should be matched (10-4 to 2000/s).

Response 5: This problem has been modified in the paper.

Point 6: The legend in Figure 3(a) should be revised. I think one of blue and green lines should be 72h-PA12.

Response 6: The error has been corrected in the article.

Point 7:  In Figure 3, the height of CF/PA12-0h was higher than pure PA12. Please explain why.

Response 7: Adding carbon fiber to pure PA12 can increase its enthalpy of melting.

Point 8:  The ‘space’ should be added between numbers and units.

Response 8: We have modified this problem in the article and marked it in yellow color.

Point 9:  It is better to make the same range of Y-axis in Figure 4 for the direct comparison of (a) and (b).

Response 9: The error has been corrected in the article.

Point 10:  There is no explanation about Figure 6 in the manuscript.

Response 10: It is observed that the slope of the curve in Fig. 6 steepens as the strain rate increases. As can be seen from Fig. 6, with the increase of strain rate, the peak stress and modulus of composite material increase significantly. Although the peak strain decreases with the increase of strain rate, the dynamic failure strain and failure stress are much lower than the quasi-static failure strain.

The above words appeared in the original article, we guess that the review experts may be missed.

Point 11: To Figure 9, the expression in figures and manuscript are different. They should be matched.

Response 11: The error has been corrected in the article.

Point 12:  In section 3.2.2., “ more than 15%” and “less than 4 MPa (8%)” should be explained how they were calculated. It will be helpful to add magnified images to explain those values.

Response 12: It was not convenient to enlarge the picture, so we marked the data on the picture.

Point 13: In section 3.3, it seemed better to place Figure 11 before Figure 10 for the better understanding of cross-sections.

Response 13: We have reversed the order in the article.

Point 14:   The purpose of section 3.3 is not clear. It is better to mention about the relationship of section 3.3 and related experiments between previous and following chapters.

Response 14: We have added relevant content to the original text.

In the quasi-static compression experiment, only CF/PA12 samples were broken, so the cross-section was photographed by SEM to reveal the cause of fracture.

Point 15: The comparison in necessary between PA12 and CF/PA12 to mention that the only CF/PA12 was fractured as authors mentioned.

Response 15: On the basis of the original figure, we added the morphology of PA12 samples.
